# Cardiorespiratory Fitness and Diet Quality Profile of the Lithuanian Team of Deaf Women’s Basketball Players

**DOI:** 10.3390/ijerph17186749

**Published:** 2020-09-16

**Authors:** Marius Baranauskas, Valerija Jablonskienė, Jonas Algis Abaravičius, Rimantas Stukas

**Affiliations:** 1Department of Physiology, Institute of Biomedical Sciences, Biochemistry, Microbiology and Laboratory Medicine of the Faculty of Medicine, Vilnius University, 01513 Vilnius, Lithuania; valerija.jablonskiene@mf.vu.lt (V.J.); algis.abaravicius@mf.vu.lt (J.A.A.); 2Department of Public Health, Institute of Health Sciences of the Faculty of Medicine, Vilnius University, 01513 Vilnius, Lithuania; rimantas.stukas@mf.vu.lt

**Keywords:** high-performance athletes, deaf athletes, athletes with disabilities, actual nutrition, body composition, cardiorespiratory fitness

## Abstract

There are about 466 million people with hearing impairments in the world. The scientific literature does not provide sufficient data on the actual nutrition and other variables of professional deaf athletes. The objectives of this study were to investigate and evaluate the body composition, the physical working capacity, the nutrition intake, and the blood parameters of iron and vitamin D in the Lithuanian high-performance deaf women’s basketball team players. The female athletes (n = 14) of the Lithuanian deaf basketball team aged 26.4 ± 4.5 years were recruited for an observational cross-sectional study. A 7-day food recall survey method was used to investigate their actual diet. The measurements of the body composition were performed using the BIA (bioelectrical impedance analysis) tetra-polar electrodes. In order to assess the cardiorespiratory and aerobic fitness levels of athletes, ergo-spirometry (on a cycle ergometer) was used to measure the peak oxygen uptake (VO_2peak_) and the physical working capacity at a heart rate of 170 beats per minute (PWC_170_). The athletes’ blood tests were taken to investigate the red blood cells, hemoglobin, 25-hydroxyvitamin D, ferritin, transferrin, iron concentrations, and total iron-binding capacity (TIBC). The consideration of the VO_2peak_ (55.9 ± 6.1 mL/min/kg of body weight, 95% CI: 51.8, 58.9) and the low VO_2peak_ (56–60 mL/min/kg of body weight) (*p* = 0.966) in the deaf women’s basketball team players revealed no differences. For the deaf female athletes, the PWC_170_ was equal to 20.3 ± 2.0 kgm/min/kg of body weight and represented only the average aerobic fitness level. The carbohydrate and protein intakes (5.0 ± 1.3 and 1.3 ± 0.3 g/kg of body weight, respectively) met only the minimum levels recommended for athletes. The fat content of the diet (38.1 ± 4.1% of energy intake) exceeded the maximum recommended content (35% of energy intake) (*p* = 0.012). The mean blood serum concentrations of 25(OH)D and ferritin (24.1 ± 6.6 nmol/L and 11.0 ± 4.1 µg/L, respectively) predicted vitamin D and iron deficits in athletes. Female athletes had an increased risk of vitamin D and iron deficiencies. Regardless of iron deficiency in the body, the better cardiorespiratory fitness of the deaf female athletes was essentially correlated with the higher skeletal muscle mass (in terms of size) (r = 0.61, *p* = 0.023), the lower percentage of body fat mass (r = −0.53, *p* = 0.049), and the reduced intake of fat (r = −0.57, *p* = 0.040).

## 1. Introduction

There are about 466 million people with hearing impairments in the world [1]. The term a deaf athlete refers to an athlete with a hearing impairment (deaf, hard of hearing) and an athlete who has a cochlear implant. In order to have the right to participate in the international elite competitions for deaf athletes, where it is forbidden to use a hearing aid or an external part of the cochlear implant during the competition, the only restriction is related to meeting the criterion of a minimum damage of 55 dB on a better ear, the average speech frequencies of 500, 1000, and 2000 Hz [2,3]. The international elite competitions for deaf athletes include major sports events such as the Deaflympics, previously called The World Games for the Deaf, The World Deaf Championships, and The Continental Deaf Championships (European and other championships) in the Olympic and non-Olympic sports.

The scientific literature provides very little data on the nutrition, body composition, and physical capacity of deaf athletes. Overall, there is no scientific evidence that identifies an association between the body composition, hematological profile, and physical capacity of the deaf athletes’ population. Basketball is a team sport with a large part of energy produced during anaerobic reactions triggered by physical loads [4]. However, when playing basketball, aerobic energy is used to convert glucose and fat into energy and to support the movements of lower intensity and longer duration which represent about 65% of the active game time [5]. In this respect, the cardiorespiratory capacity of basketball players is very important [6]. The VO_2peak_ being one of the most commonly used indicators of aerobic power and metabolism, is considered the gold standard and plays a key role in measuring the aerobic ability. A correlation was found between VO_2peak_, aerobic reactions, and the improved physical capacity during a basketball game; i.e., the higher VO_2peak_ is more effective in the performance of multiple sprints, running, and jumping during long-lasting competitions [5,7,8,9]. Even though the oxygen uptake is increasing along with an increase in the physical work capacity, the oxygen uptake has its limits. This limit depends on the ability of the respiratory and circulatory systems to provide oxygen to the working muscles. The muscles are able to pick up and uptake a certain amount of oxygen. This is determined by the number of mitochondria and the amount and activity of oxidative enzymes in the muscles. Thus, the VO_2peak_ is an indicator that directly and comprehensively shows the functional capabilities of the respiratory and circulatory systems, and the ability of muscles to maximize the uptake of oxygen. The VO_2peak_ is determined by many factors such as the maximal voluntary ventilation, the composition of blood (hemoglobin, iron concentrations), vascular elasticity, muscle capillary density, mitochondrial quantity and activity, the levels and the activity of the muscle oxidative enzymes [10,11]. The impairments of speech and communication observed in deaf people have also been shown to have a negative impact on their cardiorespiratory capacity [12,13]. A limited cardiorespiratory fitness and lower VO_2peak_ may be due to the underdeveloped lung volume resulting from the deaf people’s restricted speech, singing, or shouting [14,15]. In addition, the deaf athletes are not only sportsmen with a hearing loss sharing a unique cultural identity, but they may also exhibit some differences in the cardiac structure and the global left ventricular performance that cannot be explained only by their special disablement [16].

The optimal diet is inextricably linked with the improved levels of the overall health and fitness of athletes. The scientific evidence suggests that the athletes with disabilities (except for the deaf athletes) consume excess fat [17,18], too little of carbohydrates, and a dietary fiber [19]. The protein content in the diet of the disabled athletes is sufficient [19,20]. However, the insufficient consumption of vitamin D with food remains one of the most pressing nutritional problems in athletes, especially in the athletes with disabilities. Moreover, clinical trials show a lack of an active form of vitamin D in athletes’ serum [21,22,23]. The deficiency of iron storage in the body leads to anaemia and the deterioration of aerobic capacity of athletes [24,25]. Women athletes in particular are at risk of iron deficiency [18,25,26,27,28]. Therefore, once the iron deficiency in the body has been confirmed by clinical laboratory tests, an adjustment of the athletes’ diet with iron supplements is necessary in exceptional cases. To our knowledge, the scientific literature does not provide any data on the actual nutrition of deaf athletes. The current lack of scientific data does not make it possible to optimise and adjust the diets of deaf athletes in order to meet the nutritional goals of appropriate sports. The objectives of this study were to investigate and evaluate the body composition, physical working capacity, nutrition intake, and blood parameters of iron and vitamin D in Lithuanian high-performance deaf women’s basketball team players.

## 2. Material and Methods

### 2.1. Study Population

In May 2018, 26.4 ± 4.5 year-old Lithuanian high-performance deaf women’s basketball team players with hearing impairments were surveyed. Only the deaf female athletes (N = 14) who were taking part in the World and European Deaf Championships and were preparing for the Deaflympics were recruited for an observational cross-sectional study.

The women basketball players participated in organised physical activities for 14.6 ± 6.3 years, while workouts were done 4–6 days a week. The workouts lasted for 359.1 ± 173.8 min per week, with an average workout time of 52.2 ± 25.2 min per day. The athletes were investigated during the special preparatory period for the competition. The dimensions of the training load of the athletes were in accordance with the approved training plans. In order to evaluate the training load plans of the athletes involved in the study, the training plans for athletes officially approved by the Department of Physical Culture and Sports and the Lithuanian National Olympic Committee were followed (Table 1).

### 2.2. Anthropometric Measures

The height of the athletes was measured with a stadiometer (±1 cm) at the Lithuanian Sports Medicine Centre. The measurement of the body weight (BW) and individual weight components such as lean body mass (LBM, in kg and %), muscle mass (MM, in kg and %), and body fat (BF, in kg and %) were performed at the Lithuanian Sport Centre using the BIA (bioelectrical impedance analysis) tetra-polar electrodes (13 lot 21 block with certification EN ISO 13488; Jinryang Industrial Complex, Kyungsan City, Korea) and resistivity was measured with 8–12 tangent electrodes at different frequencies of the signal: 5, 50, 250, 550, and 1000 kHz [29]. The muscle and fat mass indexes (MFMI) of each athlete were calculated by dividing the weight of the muscle (in kg) by weight (in kg). The athletes’ MFMI was rated on a scale that described a very low MFMI (<1.8), a low MFMI (1.9–2.89), an average MFMI (3–3.99), a large MFMI (4–5), and a very high MFMI (>5) [6].

### 2.3. Energy Requirements

The basal metabolic rate (BMR), daily energy expenditure (DEE), and training energy expenditure (TEE) of all subjects were estimated. BMR was calculated using the Harris and Benedict formulas [30]. The physical activity and lifestyle variables considered in this study were a 7-day period spent on regular and non-regular activities, sedentary activities, and sleeping habits as reported by the American Dietetic Association, Dietitians of Canada, and the American College of Sports Medicine [31]. The participants recorded a 7-day period frequency and duration of physical activities and the rate of energy expenditure for each activity was estimated. These measures were supported by the studies of Ainsworth et al. [32] and the data were managed according to the specific activity. Some normative data of the basic physical activities, the activity codes, and METs (in kcal/kg/h) for physical activities were used in the study (code 15055, basketball, general, METs −6.5 kcal/kg/h; code 15711) [32]. The metabolic equivalent intensity level (MET) was used; it should be understood as the ratio of work metabolic rate to a standard resting metabolic rate of 1.0 (4.184 kJ)/kg/h (or 3.5 mL O_2_/kg/min). The physical activity level (PAL) of all the athletes was calculated as the ratio between DEE and BMR as reported by FAO/WHO/UNU [33]. In order to estimate the total energy requirement (EER), the basal metabolic rate (BMR) was then multiplied by the appropriate activity factor (PAL).

### 2.4. Dietary Intake

The actual diet of the Lithuanian high-performance athletes was estimated using a 7-day food recall survey method [34,35]. The survey was performed by a trained interviewer using the direct interview method at the Lithuanian Sports Centre. The interviewer recorded the data on food products/stuffs and meals, consumed by each athlete. In the course of the food recall, the portion sizes in the special Atlas of Foodstuffs and Dishes were used. The atlas displays different portions of products and meals assessed in grams, making it possible to record the amounts of all the food products and meals consumed [36]. The average daily food intake of athletes was evaluated, and the chemical composition and energy value of the food rations were determined using chemical composition tables [37].

The carbohydrate, protein, and fat intake were assessed according to the recommendations found in the academic literature. The recommended carbohydrate content for deaf athletes must amount to 5–8 g/kg of BW [38], and the protein content must amount to 1.4–1.6 g/kg of BW [19,20]. The percentage of energy should be between 20 to 35% from fat, <10% from saturated fatty acids (SFA), from 6 to 10% for polyunsaturated fatty acids (PUFA), from 1 to 2% for omega-3 fatty acids (omega-3 FA), and from 5 to 8% for omega-6 fatty acids (omega-6 FA) [31,39].

The increased intake of vitamins and/or minerals in athletes has little or no effect on the physical performance indicators as long as the dietary intake of vitamins and minerals meets the prescribed Recommended Dietary Intake (RDI) for a given country. Thus, athletes should consume the diets that provide at least the RDI for all micronutrients [40]. In our study, the same RDI for vitamins and minerals in the Lithuanian population [27] was also applied to examine the diets of deaf athletes in Lithuania [41].

### 2.5. Blood Collection

The blood samples were collected from the participants by a trained phlebotomist at the accredited Lithuanian Sports Medicine Centre medical laboratory (clinical laboratory license no. 2545; Ozo 39 St., Vilnius, Lithuania). To control the exercise-induced influences on the results, the participants were asked to refrain from physical activity for 24 h before blood collection and to be present in a hydrated and fasted state. The samples were analysed by a hematological analyser MEK6400-K (INHON KOHDEN with certification EN ISO 11463; Japan) for hematology profile (haemoglobin (Hb), red blood cells count (RBC)), serum iron, ferritin, total iron-binding capacity (TIBS), transferrin saturation, and 25-hydroxyvitamin D (25(OH)D)) at an accredited biomedical laboratory ANTĖJA (clinical laboratory license no. LA-173-09; Viršuliškių 65A St., Vilnius, Lithuania). The blood samples were tested in singlet, with any abnormal results repeated to verify the results. The laboratories monitored the accuracy and the precision of the samples.

### 2.6. Physical Working Capacity

The aerobic capacity (VO_2peak_) was measured in each deaf athlete using a direct ergo-spirometry. The cardiopulmonary exercise test was performed on a cycle ergometer Ergoline-Select 200 (Sensormedics with no. A8Y11425M and certification EN ISO 0459; Savi Ranch Pkwy, Yorba Linda, CA, USA) using a ramp protocol, with linear and progressive load increments until physical exhaustion and/or the onset of the limiting signs and symptoms. The protocol consisted of the incremental workloads starting at a load of 100 W and progressing by 50 W steps until volitional exhaustion. The VO_2_ uptake was recorded during the study. A continuous expired gas analysis was measured with the Vmax V229 (Ergoline GmbH with no. SN2008000977 and certification EN ISO 0123; Germany) print automated system. VO_2peak_ was defined as the mean VO_2_ (L) over the last 30 s of the test. VO_2peak_ was also adjusted for body mass (mL/kg/min). The heart rate (HR) was measured every minute during the exercise and the recovery. However, the third recovery pulse was recorded from 180 s after the cessation of the exercise. The female athletes’ VO_2peak_ was rated on a scale that describes a very low VO_2peak_ (<55 mL/min/kg of BW), a low VO_2peak_ (56–60 mL/min/kg of BW), VO_2peak_ below an average (61–65 mL/min/kg of BW), an average VO_2peak_ (66–70 mL/min/kg of BW), VO_2peak_ above an average (71–75 mL/min/kg of BW), a high VO_2peak_ (76–80 mL/min/kg of BW), and a very high VO_2peak_ (>81 mL/min/kg of BW) [6]. While the athletes were working on a cycle ergometer, their adaptation to physical loads was evaluated applying the test of the Physical (Aerobic) Working Capacity at a heart rate of 170 beats per minute (bpm) (PWC_170_). The aerobic fitness level (AFL) of female athletes was rated on a scale that describes a very low AFL (PWC_170_ < 12 kgm/kg/min of BW), a low AFL (PWC_170_ = 12–15 kgm/kg/min of BW), an average AFL (PWC_170_ = 16–20 kgm/kg/min of BW), a high AFL (PWC_170_ = 21–25 kgm/kg/min of BW), and a very high AFL (PWC_170_ > 25 kgm/kg/min of BW) [6].

### 2.7. Statistical Analysis

SPSS V.25 for Windows (Armonk, NY, USA) and Microsoft Excel (Seattle, WA, USA) were used to complete all the statistical analyses. All the normally distributed data are presented as means ± standard deviations (SD) and all the non-normally distributed data are presented as median ± standard errors (SE). The Shapiro-Wilk W-test was used to determine normality.

To avoid the bias in our research, we identified the carbohydrate intake recommendations for low, middle, and high levels (5, 6, and 8 g/kg/day), protein (1.4, 1.6, and 1.8 g/kg/day), fat (20, 25, and 35% calories), SFA (8, 10, 12% calories), PUFA (6, 8, and 10% calories), omega-3 FA (1, 1.5, 2% calories), omega-6 FA (5, 6.5, 8% calories), respectively. When normality was confirmed, the paired samples’ *t*-tests were used to assess the differences between the groups (EI vs. EER; protein, carbohydrates, fat, saturated fatty acids, polyunsaturated fatty acids, dietary fiber, vitamins, and minerals such as A, D, K, B_9_, potassium, calcium, magnesium, phosphorus, copper, iodine, chrome vs. recommendations). When the normality assumption was violated, Wilcoxon Signed Rank tests were used to assess the mean differences between the non-normally distributed variables (the mean intakes of vitamins E, C, B_1_, B_2_, B_3_, B_6_, B_5_, B_7_, B_12_, and minerals such as iron, zinc, manganese, and selenium) and RDI for vitamins and minerals. Pearson’s and Spearman’s correlations (parametric and non-parametric tests, respectively) were used to assess the relationship between the indices of physical capacity (VO_2peak_ and PWC_170_) and the variables of interest (dietary intake, body composition, serum concentrations of 25 (OH) D), TIBS, ferritin, transferrin, iron, RBC, and Hb). The ‘r’ and/or ‘rho’ values of <0.25, 0.26–0.50, 0.51–0.75, and >0.75 were considered weak, moderate, fair, and strong associations, respectively. A *p*-value of <0.05 was used to determine the statistical significance.

### 2.8. Ethics Statement

All the organisational issues regarding the survey were discussed with the Lithuanian Sports Centre and the Bioethics Committee prior to the research. The study was conducted in accordance with a permit to carry out the biomedical research issued by the Vilnius Regional Committee of Biomedical Research Ethics (no. 158200-17-898-419, of 11 April 2017). Prior to testing, all the athletes provided their written consent and the study protocols were approved by the Lithuanian Sports Medicine Center Institutional Review Board. The clinical investigations were conducted according to the principles expressed in the Declaration of Helsinki.

## 3. Results

### 3.1. Body Composition and Physical Working Capacity

As indicated in Table 2, the BW, LBM, and MM of basketball players met the norm limits set for athletes. The BF of deaf female athletes was 24.2 ± 3.8% of BW and corresponded to the optimal. In addition, MFMI was calculated. MFMI was 3.0 ± 0.7 and corresponded to the average MFMI.

The mean resting HR, the peak HR, and the recovery HR were 62.4 ± 7.5, 186.9 ± 57.3, and 87.3 ± 10.3 bpm, respectively. VO_2peak_ for the deaf female players was 55.9 ± 6.1 mL/min/kg of BW, which corresponded to a low aerobic capacity and did not differ from the low VO_2peak_ (56–60 mL/min/kg BW) (*p* = 0.966). The athletes’ PWC_170_ was equivalent to 20.3 ± 2.0 kgm/min/kg of BW and did not differ from the PWC_170_ (16–20 kgm/kg/min of BW), which represented the average level of aerobic fitness for athletes (*p* = 0.898).

The associations between the peak HR (bpm), VO_2peak_ (mL/kg/min), and the recovery HR (bpm) were r = 0.92 (*p* < 0.001) and r = 0.33 (*p* = 0.267), respectively.

### 3.2. Energy Requirements and Dietary Intake

The study revealed the factors that determine energy metabolism in deaf female athletes. Significant and non-significant differences (*p* ≤ 0.05 and *p* > 0.05) were observed between all the energy and macronutrient recommendations when compared to the actual intakes. The estimated BMR of female athletes was 1486 ± 88 kcal, and TEE was 768 ± 291 kcal. According to Table 3, the EI of athletes was 2579 ± 590 kcal and EI did not differ from the EER of 2402 ± 355 kcal (*p* = 0.178). The carbohydrate intake of the deaf basketball players was 5.0 ± 1.3 g/kg of BW and that amount corresponded only to the minimum recommended amount of 5.0 g/kg of BW (*p* = 0.967). The paired samples *t*-tests between the reported carbohydrate intake and the moderate and high recommended amounts were all statistically significant. Similarly, the paired samples *t*-tests between the reported protein intake (1.3 ± 0.3 g/kg of BW) and the moderate and high recommended amounts were all statistically significant (Table 3). The dietary protein intake of deaf athletes met only the minimum recommended protein intake for the athletes with disabilities (1.4 g/kg of BW) (*p* = 0.129). Additionally, the paired samples *t*-tests between the reported fat intake (38.1 ± 4.1% calories) and the low, and the moderate recommended amounts were found to be statistically significant (*p* < 0.001 and *p* < 0.001, respectively). The dietary fat intake among deaf female athletes was 3.1 ± 4.1% higher than the maximum recommended one (35% calories) (*p* = 0.015). Finally, deaf athletes consumed too much SFA (13.5 ± 1.2% calories) and such dietary fat intake exceeded the maximum recommended amount (10% calories) (*p* < 0.001). On the other hand, female athletes consumed too little PUFA, omega-3 FA, and omega-6 FA. The amounts of PUFAs consumed by athletes (PUFA, 4.7 ± 1.2% calories, *p* = 0.002; omega-3 FA, 0.8 ± 0.3% calories, *p* = 0.004; and omega-6 FA, 3.8 ± 1.1% calories, *p* = 0.001) were significantly below the minimum recommended levels (6, 1, and 5% calories, respectively). Furthermore, athletes also consumed too little dietary fiber, i.e., 22.6 ± 1.2 g per day. The difference between the amount of dietary fiber consumed by athletes and the minimum recommended intake of dietary fiber (25 g) was −2.9 ± 4.4 g (95% CI: −5.5, 0.3) (*p* = 0.029).

The vitamin and mineral composition in the diets of athletes was evaluated, as indicated in Table 4 and Table 5. Taking into account that some of the data of the perceived needs and the intake were not normally distributed, Wilcoxon Signed Rank tests were used to assess the differences between the perceived needs and the perceived intake as well as between the perceived intake and the actual intake. It was found that the amounts of vitamins B_1_, B_2_, B_3_, B_6_, B_9_, B_12_, C, minerals magnesium, phosphorus, copper, and iodine consumed by deaf basketball players with food and dietary supplements were significantly (*p* < 0.05) higher than the RDI. The average daily intake of vitamins A, E, K, B_5_, B_7_, minerals potassium, calcium, zinc, manganese, chromium, and selenium by deaf athletes complied with RDI of vitamins and minerals (*p* > 0.05). However, the results of our study confirmed the vitamin D and iron deficiencies in the diets of athletes. The average vitamin D intake (244 ± 228 International Unit (IU)) was significantly lower than RDI (400 IU) (−156 ± 228 IU (95% CI: −288, −24), *p* = 0.023). Similarly, the iron content in the diets of athletes (10.4 ± 1.2 mg) was lower than RDI (15 mg) (*p* = 0.050).

### 3.3. Blood Analysis

The analysis of the blood samples of deaf female athletes confirmed a lack of an active form of vitamin D (25(OH)D) in the serum. The serum concentration of 25(OH)D is the best indicator of vitamin D status. It reflects that vitamin D is produced cutaneously and obtained from food and supplements, and has a fairly long circulating half-life of 15 days. Based on our study data, the average serum 25(OH)D concentration of deaf female athletes was 24.1 ± 6.6 nmol/L (95% CI: 19.9, 28.2). A more detailed analysis of the study data confirmed a serum 25(OH)D deficit (<20 nmol/L) in almost one third (28.6%) of deaf athletes. A lack of serum 25(OH)D (20.1–30 nmol/L) was found in 57.1% of the women under analysis. A sufficient concentration of serum 25(OH)D (>30 nmol/L) was found only among 14.3% of athletes.

In addition, the analysis of blood samples confirmed a lack of iron storage in the bodies of deaf basketball players (Table 6). The serum ferritin concentration (11.0 ± 4.1 µg/L) indicating the iron storage in the bodies of athletes was found to be below the norm (15–150 µg/L). The average serum iron concentration of 8.8 ± 2.3 µmol/L corresponded to the minimum norm limit (6.6–26 µmol/L). Moreover, the analysis of 10 out of 14 blood samples confirmed a lack of RBC in the blood of female athletes. The predictive iron deficiency anemia was confirmed by the Hb level (126.3 ± 9.2 g/L) in the blood of the studied athletes, which corresponded to only the minimum norm of Hb (120–160 g/L).

### 3.4. Association between Body Composition, Haematological Profile, Dietary Intake, and Cardiorespiratory Fitness

The correlations between the VO_2peak_, the physical working capacity, the haematological parameters, and the dietary intakes are given in Table 7. Two correlations (positive and negative) between the serum transferrin (g/L), the dietary fat intake (% calories), and VO_2peak_ (mL/kg/min) were identified, while the coefficients were r = 0.59 (*p* = 0.028) and r = −0.57 (*p* = 0.040), respectively. The correlations between the VO_2peak_ and the dependent variables of interest (the parameters of body composition) are presented in Figure 1. Overall, there were negative correlations between the VO_2peak_ (mL/kg/min) and the BF (% of BW) (r = −0.53, *p* = 0.049). The high positive values of ‘r’ between VO_2peak_ (mL/kg/min) and LBM (% of BW), MM (% of BW), and MFMI were 0.56 (*p* = 0.039), 0.61 (*p* = 0.023), and 0.50 (*p* = 0.047), respectively.

## 4. Discussion

Deaf athletes are different from all other athletes because of their special communication needs on the sports field. That is why the International Olympic Committee (IOC) organises Deaflympics as a separate sports organisation for the athletes with a hearing loss [3]. This is the first study to show the cardiorespiratory fitness and nutrition profile in deaf women’s basketball team players. The limited cardiorespiratory fitness and lower VO_2peak_ may be due to the underdeveloped lung volume resulting from the restricted speech, singing, or shouting of deaf people [14,15]. Apart from loss of hearing, the deaf athletes have many cardiac differences from their normal counterparts in terms of the left ventricular structure and function [16]. The researchers point out that VO_2peak_ (about 40 mL/kg/min of BW) of deaf athletes is significantly lower than VO_2peak_ of healthy athletes [42,43]. Nevertheless, according to our study, the aerobic capacities of the deaf women’s basketball players as assessed by the peak VO_2_ (55.9 mL/min/kg of BW) were also similar to the normal athletes (50.7–51.9 kgm/min/kg of BW) [16,44]. These data suggest that the deaf women athletes from Lithuania can show the same physical performance with the same aerobic capacity and they are also not different in terms of the cardiac autonomic function. However, the VO_2peak_ (55.9 mL/min/kg of BW) of the Lithuanian deaf women’s basketball team players does not differ from the low VO_2peak_ (56–60 mL/min/kg of BW). On the other hand, it is well established that a linear relationship exists between the work heart rate and the oxygen intake at submaximal workloads, but the same relationship does not exist during recovery. The factors which govern the rate of a decrease in the heart rate during recovery are not well investigated [45]. In our study, it is shown that the recovery heart rate at the beginning of recovery was influenced only by the peak of the heart rate during work and has no association with VO_2peak_. These results mean that the myocardial efficiency and economy in deaf athletes are suboptimal and it can be improved by adapting a special training process during the macrocycle. However, we suggest that the findings in our study shall set the basis for future studies. In addition, the aerobic fitness level of the deaf basketball players under our analysis is only average, as confirmed by their PWC_170_, which equals 20.3 kgm/min/kg of BW. The comparison of the data obtained from the relevant studies of the healthy basketball players in Lithuania revealed a significantly higher PWC_170_ (23 kgm/min/kg of BW) [6].

When analysing the body composition of Lithuanian deaf athletes, we found that the body fat mass (24.2% of BW) of the deaf women we studied was similar to the body fat mass (24.50% of BW) of elite deaf athletes from Turkey [46], and was greater than the body fat mass (21.2% of BW) of female deaf athletes of the Polish deaf national team [47]. Meanwhile, the lean body mass reading (49.0 kg) in the deaf athletes we investigated in Lithuania was higher than the lean body mass dimensions in Polish and Turkish elite deaf athletes that makes up 47.7 and 45.07 kg, respectively [46,47]. In connection with these data, it can be emphasized that the body composition of Lithuanian deaf women’s basketball team players is more appropriate for high athletic performance. This hypothesis was partially confirmed by the essential relationship we found between the body composition of female athletes and the cardiorespiratory fitness. Specifically, the high values of correlation coefficients between the VO_2peak_ and the body composition parameters such as lean body mass, muscle mass, and muscle and fat mass index suggest that the muscle mass is the main factor that ensures a higher level of cardiorespiratory fitness in deaf athletes. Moreover, the VO_2peak_ and the body fat percentage in Lithuanian deaf athletes showed a negative correlation by Pearson test and was statistically significant. The results of our study are consistent with the data published by a small number of researchers who investigated the samples of athletic population and reported the negative correlation between the VO_2peak_ and the body fat percentage [48] and the significant relationship between the muscle mass and the VO_2peak_ [49].

In our study, the actual diet of the deaf women’s basketball team players was assessed for the first time. The total daily energy requirement (EER) of athletes with disabilities should vary on average within the range from 1500 to 2300 kcal [25,50]. The results of our study showed that the energy intake (EI) of deaf athletes equals 2579 kcal and corresponds to the EER (2402 kcal). The nutrients are unbalanced in the athletes’ diets due to an excessive fat consumption. More specifically, the fat intake of athletes (37% calories) is higher than the recommended one (20–35% calories) [31]. Meanwhile, the athletes with disabilities (wheelchair athletes, athletes with spinal cord injury) consume varying amounts of fat (% of calories) ranging from 29 to 44% [17,18,25]. A similar fat intake (35–39% calories) was found among the healthy NCAA Division I female athletes competing in basketball in USA [51]. Our study has also found that athletes consume too much saturated fatty acids (SFA) and too little polyunsaturated fatty acids (PUFA), omega-6 FA, and omega-3 FA. There is an epidemiological evidence that a high-fat diet promotes the development of obesity and cardiovascular risk in the inactive population group [52]. There is no scientific evidence that high-fat food intake results in overweight athletes or in increased cholesterol levels. However, high-fat diets have a correlation to homocysteine levels in the body, promoting the emergence of cardiovascular diseases [53,54]. However, a significant negative relationship between VO_2peak_ and the high-fat diet also existed among the Lithuanian deaf athletes we investigated. On the other hand, anti-inflammatory effects of omega-3 FA intake may reduce muscle damage or enhance recovery from intense, eccentric exercises [55], therefore, the use of omega-3 FA dietary supplements (about 2 g/day) is recommended for deaf athletes. The best recommendation may be to include rich sources of omega-3 fatty acids, such as fatty fish, in the diet instead of supplements. Fish oil or omega-3 FA supplement consumption could include heavy metal contaminants, or cause bleeding, digestive problems, and/or increased low density lipoproteins (LDL).

The scientific evidence shows that the athletes with disabilities consume too little carbohydrates (3.5 g/kg of BW) [19]. Carbohydrate intake is lower than the recommended one (5–10 g/kg body weight) [38]. Meanwhile, the athletes we studied consume a minimum amount of carbohydrates (5 g/kg of BW), which is enough to overcome the 52-min daily physical workloads. The deaf athletes we studied and the athletes with disabilities investigated by other authors consume too little dietary fiber (<25 g/day) as a form of carbohydrate that is not broken down during digestion [19,25,50]. Purposefully to prevent gastrointestinal disorders (e.g., constipation) and to improve physical performance [56], the athletes with disabilities are recommended to consume a diet higher in dietary fiber with a sufficient amount of liquids [57].

For healthy professional athletes a protein intake of 1.4–2.0 g/kg of BW is recommended [58]. Protein intake recommendations for athletes with disabilities are different and correspond to 1.4–1.6 g/kg of BW per day [19,20]. In contrast to Paralympic athletes from Canada and Spain [19,20], the deaf female athletes we studied in Lithuania consume too little protein (1.3 g/kg of BW). The protein intake mentioned above meets only the minimum recommended level (1.4 g/kg of BW) [58]. Basketball is a team sport where during physical loads a large part of energy is produced by anaerobic reactions. Therefore, basketball players are advised to ensure muscular hypertrophy, muscular strength, and anaerobic power in the specific training process. An inadequate intake of protein and essential amino acids (especially leucine) can lead to the negative nitrogen balance, the muscle and the whole body protein catabolism, slower skeletal muscle adaptation to training and performance [58]. Therefore, the nutrient imbalance in the diet of the Lithuanian national women’s deaf basketball team players due to excessive fat intake should be corrected. High-performance deaf athletes are recommended to increase their protein intake while reducing fat consumption.

The athletes must consume RDI-compliant amounts of vitamins and minerals with food [40]. According to our study results, vitamin D deficiency and iron deficiency was found in the diet and body of the deaf women athletes. Similarly, low serum vitamin D levels have been found in healthy and disabled athletes in other countries such as Canada, Swiss, and Ireland [21,22,23]. It should be noted that the serum concentration of 25(OH)D is the best indicator of vitamin D status. It reflects vitamin D produced cutaneously and that obtained from food and supplements [59] and has a fairly long circulating half-life of 15 days [60]. Many cells have vitamin D receptors, and some convert 25(OH)D to 1,25(OH)_2_ D. Based on its review of the data of vitamin D needs, a National Academy of Medicine (IOM) concluded that persons are at risk of vitamin D deficiency at serum 25(OH)D concentrations <30 nmol/L. Practically all people are sufficient at levels ≥50 nmol/L; the committee has stated that 50 nmol/L is the serum 25(OH)D level that covers the needs of 97.5% of the population. The average serum 25(OH)D concentration of our athletes we studied is 24.1 ± 6.6 nmol/L and it does not reach the minimum rate among 90% of the deaf women’s basketball players in Lithuania. Vitamin D deficiency in the body can adversely affect the bone tissue metabolism and structure, and increase the risk of osteoporosis in athletes with disabilities [61]. Many genes encoding proteins that regulate cell proliferation, differentiation, and apoptosis are modulated in part by vitamin D [59]. Thus, vitamin D has other roles in the body, including the modulation of cell growth, the neuromuscular and immune function, and the reduction of inflammation [59,62,63]. Vitamin D has been shown to be important in the maintenance of the immune system, protein synthesis, and muscular and cardiovascular functions in athletes [64]. In accordance with the recommendations of the IOC, supplementation of between 800 and 1000–2000 IU/day is recommended to maintain the status for the general population. Supplementation guidelines have not yet been established for athletes. A short-term, high-dose supplementation which includes 50,000 IU/week for 8–16 weeks or 10,000 IU/day for several weeks may be appropriate for restoring the status in deficient athletes. Careful monitoring is necessary to avoid toxicity [55,65].

Iron deficiency, with or without anemia, can impair the muscle function and limit work capacity leading to a compromised training adaptation and athletic performance [24,25]. When comparing our research data with other scientific findings, it can be stated that the correlations between the VO_2peak_ and the haematology profile have not been clarified due to the presence of decreased serum iron, Hb, and RBC levels in almost all women deaf athletes. Overall, the high value of the positive correlation coefficient between the VO_2peak_ and the transferrin concentration in deaf athletes’ serum also suggests that the transport of iron into the bone marrow is activated due to the increased production of transferrin (in term elevated and/or undiminished transferrin level) and thus a sufficient iron content required for erythropoiesis is derived. Thus, the relationship between the VO_2peak_ and the elevated transferrin level may be explained via a compensatory pathway in which the production of transferrin is activated by a decrease in iron storage in the body [66].

The suboptimal iron status may result from the limited iron intake, poor bioavailability and/or inadequate energy intake, or an excess iron need due to a rapid growth, high-altitude training, menstrual blood loss, foot-strike haemolysis, or excess losses in sweat, urine or faeces [27,28]. Iron requirements for all female athletes may be increased by up to 70% of the estimated average requirement [31]. According to our study, the deaf female athletes have a lack of iron in their diet. The average iron intake of athletes is 11.7 mg/day and is below the recommended levels of RDI and IOC (15 and >18 mg/day, respectively) [41,55]. The iron storage in the body of the Lithuanian women’s deaf team basketball players is also insufficient at a critically low serum ferritin concentration. There is no agreement on the serum ferritin level that corresponds to a problematic level of iron depletion/deficiency, with various suggestions ranging from <10 to <35 ng/mL [31]. The average ferritin concentration (11 ± 4.1 ng/mL) of the players we studied is below the ferritin limits mentioned above. In addition, low rates of RBC and hemoglobin indicate an increased risk of anemia among the deaf basketball players in Lithuania. Similar data are also provided by other researchers on iron deficiency between female athletes (Landahl G et al. [67], Parnell JA. et al. [68], Sandström G et al. [28]) and athletes with disabilities (Eskici G et al. [18], Krempien JL and Barr SI [25], Rowland T et al. [26]). The deaf athletes who do not maintain an adequate iron status may need supplemental iron at doses greater than their RDI (i.e., >18 mg/day for women). Regardless of the etiology, a compromised iron status can negatively impact athletes’ health, physical, and mental performance, also the athletes with iron deficiency require clinical follow-up, which may include supplementation with larger doses of oral iron supplementation along with the improved dietary iron intake [31].

The limitation of our study was that the measurements of body composition (BW, LBM, MM, BF) were performed using the BIA instead of the dual-energy x-ray absorptiometry (DXA), while the DXA and anthropometry (specific Durnin and Womersley or Deborah Kerr formula) have high levels of consistency, with low bias ranges [69]. Another limitation was an observational study with a low sample size. A larger number of samples will definitely clear the results. The inclusion criteria of the study, such as the different sports represented by deaf athletes, would allow the comparison of the data. However, due to the extremely small number of high performance athletes with hearing impairments, we only studied the deaf female basketball players and this is an essential limitation of our research.

## 5. Conclusions

The VO_2peak_ and the cardiorespiratory fitness of the Lithuanian high-performance deaf women’s basketball team players are low and the aerobic fitness level is average.

The diet of deaf women athletes does not meet the requirements because it contains too much dietary fat and saturated fatty acids. For the first time, the insufficient intake of polyunsaturated fatty acids, omega-6 and omega-3 fatty acids, and dietary fiber was found in the deaf female athletes’ population. In preparation for the Deaf Europe and World Championships, the Deaflympics, the diets of deaf female athletes must be adjusted and optimised to meet the sports nutritional goals. The high-performance deaf women’s basketball team players have an increased risk of vitamin D and iron deficiencies. To avoid malnutrition, the deaf female athletes are recommended to reduce their dietary fat intake, and increase their consumption of protein, omega-3 fatty acids, and dietary fiber. The deaf women athletes are critically recommended to use vitamin D and iron supplements when their serum concentrations of red blood cells, ferritin, iron, hemoglobin, and 25-hydroxyvitamin D are at low levels.

The intercorrelations among the muscle mass, the dietary fat intake, and the VO_2peak_ of athletes were determined. Regardless of iron deficiency in the body, the better cardiorespiratory fitness of women deaf athletes is essentially associated with the higher skeletal muscle mass (in terms of size) (r = 0.61, *p* = 0.023), the lower percentage of body fat mass (r = −0.53, *p* = 0.049), and the reduced intake of fat (r = −0.57, *p* = 0.040).

## Figures and Tables

**Figure 1 ijerph-17-06749-f001:**
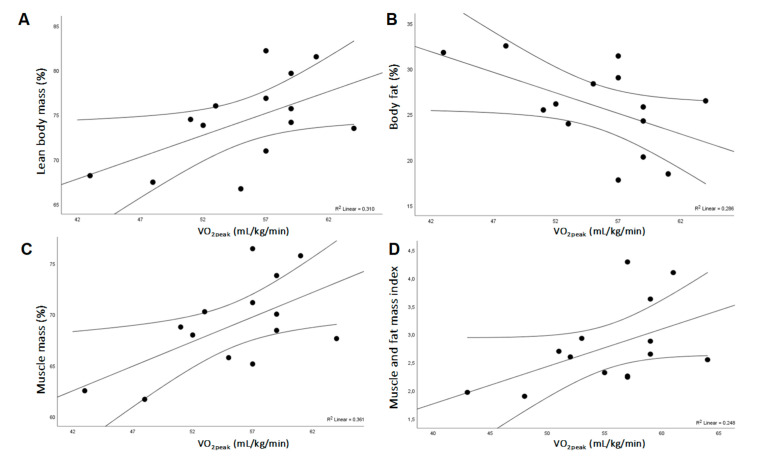
Relationship between the cardiorespiratory fitness and the body composition. (**A**)—The relationship between VO_2peak_ (mL/kg/min) and lean body mass (% of BW) (r = 0.56, *p* = 0.039); (**B**)—the relationship between VO_2peak_ (mL/kg/min) and body fat (% of BW) (r = −0.53, *p* = 0.049); (**C**)—the relationship between VO_2peak_ (mL/kg/min) and muscle mass (% of BW) (r = 0.61, *p* = 0.023); (**D**)—the relationship between VO_2peak_ (mL/kg/min) and muscle and fat mass index (r = 0.50, *p* = 0.047).

**Table 1 ijerph-17-06749-t001:** Training plans for deaf athletes.

Sports	Basketball
Athlete testing month	June
Preparatory phase of training	Special preparatory
Stage	Special training
Days of exercise per month	27
Total physical training, hours per month	40
Physical activity is divided into five intensity zones according to energy production in the muscles	
Aerobic strength endurance, recovery (blood lactate concentration up to 2 mmol/L, heart rate −130 ± 10 bpm).	17% ^1^
Aerobic strength training (blood lactate concentration is 2–4 mmol/L, heart rate −150 ± 10 bpm), special muscle-power increase at the anaerobic threshold.	41% ^1^
Mixed aerobic and anaerobic glycolytic strength training (blood lactate concentration is 4–12 mmol/L, heart rate −170 ± 10 bpm), VO_2_max increase.	34% ^1^
Anaerobic glycolytic strength training (blood lactate concentration up to 21 mmol/L, heart rate ≥181 bpm).	6% ^1^
Anaerobic phosphocreatine (maximum effort) strength training (blood lactate concentration is 1.5–6 mmol/L).	2% ^1^

^1^—Time allocated for intensity zones during workouts (%).

**Table 2 ijerph-17-06749-t002:** Body composition and physical capacity (peak oxygen uptake (VO_2peak_) and physical working capacity (PWC_170_)) of deaf athletes.

Body Composition	Mean ± SD	(95% CI)	Normative
Height (cm)	171.9 ± 6.2	(168.8, 176.3)	
BW (kg)	65.2 ± 7.8	(61.7, 72.5)	63.2 ± 5.9 ^1^
LBM (kg)	49.0 ± 5.1	(46.3, 53.1)	
LBM (% of BW)	75.4 ± 4.5	(71.3, 77.5)	70–80
MM (kg)	45.5 ± 4.4	(43.1, 49.0)	
MM (% of BW)	70.0 ± 3.8	(66.1, 71.8)	64–80
BF (kg)	15.9 ± 3.7	(14.6, 20.7)	
BF (% of BW)	24.2 ± 3.8	(22.9, 28.9)	20–24
MFMI	3.0 ± 0.7	(2.3, 3.2)	3–3.99
PWC_170_ (kgm/min/kg of BW)	20.3 ± 2.0	(18.2, 20.9)	
Resting HR (bpm)	62.4 ± 7.5	(57.2, 67.5)	
VO_2peak_ (L)	3.7 ± 0.4	(3.4, 3.9)	
VO_2peak_ (mL/kg/min)	55.9 ± 6.1	(51.8, 58.9)	
Peak HR (bpm)	186.9 ± 57.3	(150.2, 223.5)	
Recovery HR 180 s (bpm)	87.3 ± 10.3	(80.3, 94.3)	

BW: Body Weight; LBM: Lean Body Mass; MM: Muscle Mass; MFMI: Muscle and Fat Mass Index; BF: Body Fat; ^1^: Ideal Body Weight, HR: Heart Rate.

**Table 3 ijerph-17-06749-t003:** Dietary intake of deaf athletes.

EI and Nutrients	Intake Mean ± SD	Recommended ^1^		∆ Intake (Actual − Recommended)	*p*
EI (kcal)	2579 ± 590 (2202, 2956)	2402 ± 355 (2175, 2630)		176 ± 464 (−91, 444)	0.178
CHO (g/kg BW)	5.0 ± 1.3 (4.1, 5.8)	Low	5.0	0.01 ± 1.3 (−0.7, 0.7)	0.967
		Medium	6.0	−1.0 ± 1.3 (−1.7, −0.3)	0.010
		High	8.0	−3.0 ± 1.3 (−3.7, −2.3)	<0.001
PRO (g/kg BW)	1.3 ± 0.3 (1.1, 1.5)	Low	1.4	−0.1 ± 0.3 (−0.3, 0.04)	0.129
		Medium	1.6	−0.3 ± 0.3 (−0.5, −0.2)	0.001
		High	1.8	−0.5 ± 0.3 (−0.7, −0.4)	<0.001
FAT (% of EI)	38.1 ± 4.1 (35.5, 40.7)	Low	20	18.1 ± 4.1 (15.7, 20.4)	<0.001
		Medium	25	13.1 ± 4.1 (10.7, 15.4)	<0.001
		High	35	3.1 ± 4.1 (0.7, 5.4)	0.015
SFA (% of EI)	13.5 ± 1.2 (12.7, 14.3)	Low	8	5.5 ± 1.2 (4.8, 6.2)	<0.001
		Medium	10	3.5 ± 1.2 (2.8, 4.2)	<0.001
		High	12	1.5 ± 1.2 (0.8, 2.2)	0.001
PUFA (% of EI)	4.7 ± 1.2 (3.9, 5.5)	Low	6	−1.3 ± 1.2 (−2.0, −0.6)	0.002
		Medium	8	−3.3 ± 1.2 (−4.0, −2.6)	<0.001
		High	10	−5.3 ± 1.2 (−6.0, −4.6)	<0.001
Omega-3 FA (% of EI)	0.8 ± 0.3 (0.6, 0.9)	Low	1	−0.2 ± 0.3 (−0.4, −0.1)	0.004
		Medium	1.5	−0.7 ± 0.3 (−0.9, −0.6)	<0.001
		High	2	−1.2 ± 0.3 (−1.4, −1.1)	<0.001
Omega-6 FA (% of EI)	3.8 ± 1.1 (3.2, 4.5)	Low	5	−1.2 ± 1.1 (−1.8, −0.6)	0.001
		Medium	6.5	−2.7 ± 1.1 (−3.3, −2.1)	<0.001
		High	8	−4.2 ± 1.1 (−4.8, −3.6)	<0.001
Dietary fiber (g)	22.6 ± 1.2 (19.2, 24.9)	Low	20	2.1 ± 4.4 (−0.5, 4.7)	0.100
		Medium	25	−2.9 ± 4.4 (−5.5, 0.3)	0.029
		High	30	−7.9 ± 4.4 (−10.5, −5.3)	<0.001

^1^—Recommended values are derived through a combination of published review articles [19,20,31,38] and clinical experience; ∆ Intake: Actual intake–Recommended intake; all variables exhibited normal distributions using Shapiro-Wilk *W*-tests (*p* > 0.05); data are presented as means ± SD with the 95% confidence interval (CI) presented in parentheses below the mean ± SD; EI: Energy intake; EER: Estimated Energy Requirement; PRO: Protein; CHO: Carbohydrates; FA: Fatty Acids; SFA: Saturated Fatty Acids; PUFA: Polyunsaturated Fatty Acids.

**Table 4 ijerph-17-06749-t004:** Vitamin intake of deaf athletes.

Vitamins	Intake	RDI	∆ Intake (Actual − RDI)	*p*
A (µg RE ^1^)	1103 ± 876 (597, 1609) ^a^	700	403 ± 876 (−102, 909) ^a^	0.109 ^‡^
D (IU)	244 ± 228 (112, 376) ^a^	400	−156 ± 228 (−288, −24) ^a^	0.023 ^‡^
E (mg a-TE ^2^)	9.1 ± 3.3 (7.4, 21.5) ^b^	10	1.3 ± 3.3 (−3.4, 11.2) ^b^	0.363 *
K (µg)	67.8 ± 32.4 (49.1, 86.5) ^a^	75	−7.2 ± 32.4 (−25.9, 11.5) ^a^	0.422 ^‡^
B_1_ (mg)	1.7 ± 1.8 (0.3, 8.2) ^b^	1.1	0.7 ± 1.8 (−0.3, 7.5) ^b^	0.002 *
B_2_ (mg)	2.0 ± 2.0 (0.4, 9.1) ^b^	1.3	0.7 ± 2.0 (0.3, 9.2) ^b^	0.004 *
B_3_ (mg NE ^3^)	18.7 ± 12.1 (7.3, 59.6) ^b^	15	5.0 ± 12.1 (0.6, 27.6) ^b^	0.030 *
B_6_ (mg)	2.1 ± 2.8 (1.3, 13.5) ^b^	1.3	1.4 ± 2.8 (0.5, 12.1) ^b^	0.001 *
B_5_ (mg)	4.8 ± 4.6 (2.0, 21.9) ^b^	6	−0.5 ± 4.6 (−1.4, 20.1) ^b^	0.730 *
B_7_ (µg)	23.6 ± 7.3 (18.6, 50.3) ^b^	50	−24.6 ± 7.3 (−30.7, 3.7) ^b^	0.086 *
B_9_ (µg)	409 ± 166.8 (312.8, 505.4) ^a^	200	209.1 ± 166.8 (112.8, 305.4) ^a^	<0.001 ^‡^
B_12_ (µg)	5.5 ± 2.5 (4.1, 14.9) ^b^	3	4.4 ± 2.5 (1.1, 13.4) ^b^	0.008 *
C (mg)	174.0 ± 41.4 (138.9, 317.8) ^b^	80	110.4 ± 41.4 (54.8, 253.9) ^b^	0.001 *

^a^—Data are normally distributed and presented as means ± standard deviation (SD) with the 95% confidence interval (CI) in parentheses; ^b^—data are non-normally distributed and presented as medians ± standard error (SE) with the 95% confidence interval (CI) of medians in parentheses; ∆ Intake: Actual intake—Recommended intake; *—*p*-value from the Wilcoxon Signed Rank Test (non-normally distributed variables); ^‡^—*p*-value from the paired samples *t*-test (normally distributed data); ^1^—RE (Retinol Equivalent); ^2^—TE (α-Tocopherol Equivalent); ^3^—NE (Niacin Equivalent) (NE); RDI: Recommended Dietary Intake.

**Table 5 ijerph-17-06749-t005:** Mineral intake of deaf athletes.

Minerals	Intake	RDI	∆ Intake (Actual − RDI)	*p*
K (mg)	3321.7 ± 670.3 (2934.7, 3708.7) ^a^	3100	221.7 ± 670.3 (−165.3, 608.7) ^a^	0.238 ^‡^
Ca (mg)	1026.2 ± 383.0 (805.0, 1247.3) ^a^	900	126.2 ± 383.0 (−95.0, 347.3) ^a^	0.240 ^‡^
Mg (mg)	430.2 ± 191.1 (319.3, 541.1) ^a^	300	130.2 ± 51.3 (19.3, 241.1) ^a^	0.025 ^‡^
P (mg)	1349.1 ± 333.5 (1156.6, 1541.7) ^a^	700	649.1 ± 333.5 (456.6, 841.7) ^a^	<0.001 ^‡^
Fe (mg)	10.4 ± 1.2 (9.0, 21.0) ^b^	15	−3.3 ± 2.8 (−5.1, 4.4) ^b^	0.050 *
Cu (mg)	3.9 ± 2.1 (2.7, 5.1) ^a^	1	2.9 ± 2.1 (1.7, 4.1) ^a^	<0.001 ^‡^
Zn (mg)	11.2 ± 2.7 (9.9, 20.9) ^b^	10	3.5 ± 2.6 (0.1, 9.4) ^b^	0.069 *
J (µg)	201.4 ± 79.0 (155.7, 247.1) ^a^	150	51.4 ± 79.1 (5.7, 97.1) ^a^	<0.001 ^‡^
Mn (mg)	3.4 ± 0.8 (2.7, 6.0) ^b^	3	0.6 ± 0.8 (−0.1, 2.6) ^b^	0.090 *
Cr (mg)	27.0 ± 8.6 (21.7, 8.6) ^a^	40	0.2 ± 32.1 (−18.3, 18.7) ^a^	0.379 ^‡^
Se (µg)	38.0 ± 13.5 (26.1, 84.5) ^b^	50	−7.0 ± 13.5 (−19.5, 13.3) ^b^	0.551 *

^a^—Data are normally distributed and presented as means ± standard deviation (SD) with the 95% confidence interval (CI) in parentheses; ^b^—Data are non-normally distributed and presented as medians ± standard error (SE) with the 95% confidence interval (CI) of medians in parentheses; ∆ Intake: Actual intake—Recommended intake; *—*p*-value from the Wilcoxon Signed Rank Test (non-normally distributed variables); ^‡^—*p*-value from the paired samples *t*-test (normally distributed data); K: Potassium; Ca: Calcium; Mg: Magnesium; P: Phosphorus; Fe: Iron; Cu: Copper; Zn: Zinc; J: Iodine; Mn: Manganese; Cr: Chrome; Se: Selenium.

**Table 6 ijerph-17-06749-t006:** Haematology profile and serum ferritin, transferrin, iron, and TIBS concentrations in deaf athletes.

Blood Levels	Value	(95% CI)	[Min–Max]	Norm	No. of Cases below the Norm
TIBS (µmol/L)	58.4 ± 4.4 ^a^	(54.1, 60.1)	[51.3–65.5]	40.8–76.6	0
Ferritin (µg/L)	11.0 ± 4.1 ^b^	(6.8, 37.5)	[3.7–48.4]	15–150	8
Transferrin (g/L)	2.7 ± 0.3 ^a^	(2.4, 2.8)	[2.3–3.3]	2–4	0
Iron (µmol/L)	8.8 ± 2.3 ^b^	(7.4, 16.8)	[4.0–27.9]	6.6–26	3
RBC (10^12^/L)	4.3 ± 0.3 ^a^	(4.1, 4.5)	[3.6–4.8]	4.5–5.2	10
Hb (g/L)	126.3 ± 9.2 ^a^	(120.2, 132.1)	[114–143]	120–160	3

^a^—Data are normally distributed and presented as means ± standard deviation (SD); ^b^—Data are normally distributed and presented as medians ± standard error (SE); Min: Minimum; Max: Maximum; TIBS: Total Iron-Binding Capacity; RBC: Red Blood Cells Count; Hb: Hemoglobin.

**Table 7 ijerph-17-06749-t007:** Correlation between haematology profile, dietary intake, and physical capacity (VO_2peak_ and PWC_170_).

Variables	VO_2peak_ (mL/kg/min) (*p*)	PWC_170_ (kgm/min/kg of BW) (*p*)
TIBS (µmol/L) ^a^	−0.19 (0.507)	−0.45 (0.109)
Ferritin (µg/L) ^b^	−0.39 (0.171)	−0.12 (0.692)
Iron (µmol/L) ^b^	−0.06 (0.83)	−0.42 (0.130)
RBC (·10^12^/L) ^a^	−0.20 (0.497)	−0.37 (0.194)
Hb (g/L) ^a^	0.03 (0.914)	−0.29 (0.312)
Transferrin (g/L) ^a^	0.59 (0.028)	−0.05 (0.872)
TIBS (µmol/L) ^a^	0.18 (0.532)	−0.47 (0.092)
25 (OH) D (nmol/L) ^a^	−0.31 (0.287)	−0.35 (0.069)
EI (kcal/kg) ^a^	−0.13 (0.662)	0.04 (0.900)
CHO (g/kg BW) ^a^	0.07 (0.812)	0.04 (0.887)
FAT (% of EI) ^a^	−0.57 (0.040)	−0.16 (0.580)
PRO (g/kg BW) ^a^	−0.03 (0.915)	0.26 (0.363)

^a^—Data are normally distributed and presented as matrix of ‘r’ correlation coefficients; ^b^—Data are non-normally distributed and presented as matrix of ‘rho’ correlation coefficients; TIBS: Total Iron-Binding Capacity; RBC: Red Blood Cells Count; Hb: Hemoglobin; 25 (OH) D: 25-hydroxyvitamin D; EI: Energy Intake; PRO: Protein; CHO: Carbohydrates.

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
