# Peer review of "Cardiorespiratory Fitness and Diet Quality Profile of the Lithuanian Team of Deaf Women’s Basketball Players"

_ijerph, 2020, doi:10.3390/ijerph17186749_

Round 1

Reviewer 1 Report

Well written.  To my knowledge there is not much out there dealing with deaf basketball players fitness.  The authors do a good job with therirtats and explaining them.  Because of the study, there is a narrow field of people that will find this manuscript useful.  One suggestion, is for the authors to think of the results affecting other para athletes.  Other than that, good study.

Author Response

Responses to the observations provided by the reviewers

We are grateful to all reviewers for their observations and comments

Observations by Reviewer 1

Reviewer’s observations: Well written.  To my knowledge there is not much out there dealing with deaf basketball players fitness.  The authors do a good job with therirtats and explaining them.  Because of the study, there is a narrow field of people that will find this manuscript useful.  One suggestion, is for the authors to think of the results affecting other para athletes.  Other than that, good study.

Reply and corrections. We are thankful to the reviewer for a positive response. We do intend to use the results obtained from our study in practice to adjust and improve the training of para-athletes for the Paralympic Games. Taking into account a close cooperation between Vilnius University and the Lithuanian Sports Medicine Center (LSMC), the data of this study will be used to improve the monitoring of health of not only the Olympic team, but also the Paralympic team.

Observations by Reviewer 2

Reviewer’s observations: 27 - For female athletes, Is this all female athletes or just deaf athletes? line 26 and 34 clarify that you are discussing the differences.

Reply and corrections. Taking into account the reviewer's comments, we corrected line 27 in the article as follows: “For deaf athletes the PWC170 was equal to 20.3 ± 2.0 kgm/min/kg of body weight and represented only the average aerobic fitness level.”

We have also corrected sentences on lines 26 and 34 accordingly:

  1. A) „The consideration of the VO2peak (55.9 ± 6.1 mL/min/kg of body weight, 95% CI: 51.8, 58.9) and the low VO2peak (56-60 mL/min/kg of body weight) (p = 0.966) in the deaf women’s basketball team players revealed no differences.“
  2. B) „Regardless of iron deficiency in the body, the better cardiorespiratory fitness of the deaf female athletes was essentially correlated with the higher skeletal muscle mass (in term of size) (r = 0.61, p = 0.023), the lower percentage of body fat mass (r = – 0.53, p = 0.049) and the reduced intake of fat (r = – 0.57, p = 0.040).“

Reviewer’s observations: 86 - starts with an apostrophe. 

Reply and corrections. The sentence was corrected, an apostrophe deleted.

Reviewer’s observations: 110 - there are an excessive amount of parenthesis in section 2.2.

Reply and corrections. Taking into account the reviewer’s comments, we reduced the number of parentheses in Section 2.2.

Reviewer’s observations: 353 - remove "the".

Reply and corrections. Line 353 was corrected following the comments.

Observations by Reviewer 3

Reviewer’s observations: This paper is valuable as a report on the cardiorespiratory fitness and nutrition profile of deaf women's basketball team players. Although the sample size is small and there is a bias in the measurement method, it has important information that suggests that a nutritionally balanced diet will be needed to improve physical and athletic performance. Here are some comments to help authors publish successfully.

[Minor comments]

(Line 227-240; 247-248; 309-314) Where is the basis for the data described (related figures, tables or references)?

Line 227-240

“As indicated in Table 2, the athletes' height was 171.9 ± 6.2 cm. The BW (65.2 ± 7.8 kg), LBM (75.4 ± 4.5% of BW) of basketball players met the norm limits set for athletes. A more detailed analysis of the subjects' body composition showed that MM (70.0 ± 3.8% of body weight) was within the normal range. The BF of female athletes was 24.2 ± 3.8% of BW and corresponded to the optimal (20-24% of BW). In addition, MFMI was calculated. MFMI was 3.0 ± 0.7 and corresponded to average MFMI (3.0-3.9).

The mean resting HR, the peak HR and the recovery HR were 62.4 ± 7.5, 186.9 ± 57.3 and 87.3 ± 10.3 bpm, respectively. VO2peak for the deaf female players was 55.9 ± 6.1 mL/min/kg of BW, which corresponded to a low aerobic capacity and did not differ from low VO2peak (56-60 mL/min/kg BW) (p = 0.966). The athletes' PWC170 was equivalent to 20.3 ± 2.0 kgm/min/kg of BW and did not differ from the PWC170 (16-20 kgm/kg/min of BW), which represented the average level of aerobic fitness for athletes (p = 0.898).

The associations between the peak HR (bpm), VO2peak (mL/kg/min) and the recovery HR (bpm) were r = 0.92 (p < 0.001) and r = 0.33 (p = 0.267), respectively.”

Reply and corrections. In order to avoid redundant information, in this section we have reduced the number of figures that repeat the information in tables. We did not plot separate diagrams to show the correlation coefficients between peak HR (bpm), VO2peak (mL/kg/min), and the recovery HR (bpm). However, the data presented in the text are supposed to be informative enough.

Reviewer’s observations: Line 247-248

“The estimated BMR of female athletes was 1486 ± 88 kcal, and TEE was 768 ± 291 kcal. According to Table 3, the EI of athletes was 2579 ± 590 kcal and EI did not differ from the EER of 2402 ± 355 kcal (p = 0.178).”

Reply and corrections. We did not present values such as BMR, TEE, and EER in the tables, but presented the calculated values only in the text. With the tables as large as they are, we have aimed to restrict, as much as possible, the duplication of data with the information provided in the text of the article.

Reviewer’s observations: Line 309-314

“Based on our study data, the average serum 25 (OH) D concentration of deaf athletes was 24.1 ± 6.6 nmol/L (95% CI: 19.9, 28.2). A more detailed analysis of the study data confirmed a serum 25 (OH) D deficit (< 20 nmol/L) in almost one third (28.6%) of deaf athletes. A lack of serum 25 (OH) D (20.1–30 nmol/L) was found in 57.1% of the women under analysis. A sufficient concentration of serum 25 (OH) D (> 30 nmol/L) was found only among 14.3% of athletes.”

Reply and corrections. We calculated the mean serum concentration of vitamin D and divided the deaf basketball players according to the serum active vitamin D concentration into three groups (%) according to the norms. We did not provide tables with the data separately to avoid the duplication of the data in the text.

Reviewer’s observations: (Line 260-262) Isn't the maximum recommended amount of SFA 12% calories?

Reply and corrections. As indicated in the Methods section, the recommended amount of SFA is < 10% from saturated fatty acids (SFA). However, when we compared the actual SFA intake (13.5 ± 1.2) with the recommended amount, we formed a scale of 8% (low), 10% (medium), 12% (high), respectively.

Reviewer’s observations: (Line 280-282) Is the copper missing?

Reply and corrections. We are thankful to the reviewer for the observation, indeed, the deaf basketball players consume a statistically significant increase in the amount of copper compared to the recommended amount.

Reviewer’s observations: (Table 2.) The average MFMI (3.0-3.9) is missing.

Reply and corrections. Taking into account the reviewer's observation, we corrected the mentioned MFMI value in the table.

Observations by Reviewer 4

Reviewer’s observations: This article defines some performances variables and the diet of deaf women basketball players. The study is interesting because it offers new information. This topic still has to be deeply investigated in order to obtain better conclusions and more data to be compared.I just have some concerns and questions for the authors regarding the paper.

I suggest changing the title of the article from "Cardiorespiratory Fitness and Diet Quality Profile of the Deaf Women’s Basketball Team Players" to "Cardiorespiratory Fitness and Diet Quality Profile of a Deaf Women’s Basketball Team Players" or to indicate that the study was performed on the Lithuanian team.

Reply and corrections. We took into account the reviewer‘s observation and added „Lithuanian“ to the title, because the team under our analysis is the only team of the kind in Lithuania.

Reviewer’s observations: 25. What do you mean by "low VO2peak (56-60ml/Kg/min)"?

Reply and corrections. Low VO2peak shows the low aerobic powers of the subjects. In Lithuania, there are VO2peak norms set by scientists for high performance athletes (separately for men and women), according to which cardiorespiratory fitness is evaluated. There are no separate VO2peak norms for athletes of the Lithuanian national deaf team. Therefore, we evaluated the VO2peak of female athletes according to the existing standards for high-performance athletes (exclusively women). We have specified these standards in the Methods section of the article as follows:

“The female athletes’ VO2peak was rated on a scale that describes a very low VO2peak (<55 mL/min/kg of BW), a low VO2peak (56–60 mL/min/kg of BW), VO2peak below an average (61–65 mL/min/kg of BW), an average VO2peak (66–70 mL/min/kg of BW), VO2peak above an average (71-75 mL/min/kg of BW), a high VO2peak (76–80 mL/min/kg of BW) and a very high VO2peak (> 81 mL/min/kg of BW) [6].”

Reviewer’s observations: 61. during a basketball game.

Reply and corrections. We have corrected the text according to the reviewer‘s observation.

Reviewer’s observations: ENG-98-99. Change to past sentence (i.e. take-took; prepare-prepared).

Reply and corrections. We have corrected the text according to the reviewer‘s observation: „Only the deaf female athletes (N = 14) who were taking part in the World and European Deaf Championships and were preparing for the Deaflympics were recruited for an observational cross-sectional study.“

Reviewer’s observations: What do the numbers mean in the section "the content of the special work" in table 1? Please indicate.

Reply and corrections. We have clarified the meaning to "Physical exercise according to energy production in muscles is divided into five intensity zones." Also, we have clarified in the article what the numbers mean in "1  - Time for training intensity zones (%)" .

Reviewer’s observations: 195. the sentence "Significant differences were observed between the actual consumption of energy, macronutrients, and micronutrients and the recommended amounts" should not be here. Results maybe?

Reply and corrections. We have corrected (to be more exact, refused) the sentence according to the reviewer's observations.

Reviewer’s observations: 227-240 Much of the information provided here already appears in table 2. The data does not have to be presented twice, only once.

Reply and corrections. Taking into account the reviewer's observation, we refused part of the information in the Results section. We have deleted the data (numbers) on athletes' height, body weight, muscle mass, and deleted the numerical expression of body fat mass. However, we left data on cardiac output (HR, peak HR, VO2peak, PWC170).

“As indicated in Table 2, the BW, LBM, and MM of basketball players met the norm limits set for athletes. The BF of female athletes was 24.2 ± 3.8% of BW and corresponded to the optimal. In addition, MFMI was calculated. MFMI was 3.0 ± 0.7 and corresponded to the average MFMI.

The mean resting HR, the peak HR and the recovery HR were 62.4 ± 7.5, 186.9 ± 57.3 and 87.3 ± 10.3 bpm, respectively. VO2peak for the deaf female players was 55.9 ± 6.1 mL/min/kg of BW, which corresponded to a low aerobic capacity and did not differ from a low VO2peak (56-60 mL/min/kg BW) (p = 0.966). The athletes' PWC170 was equivalent to 20.3 ± 2.0 kgm/min/kg of BW and did not differ from the PWC170 (16-20 kgm/kg/min of BW), which represented the average level of aerobic fitness for athletes (p = 0.898).”

Reviewer 2 Report

 27 - For female athletes, Is this all female athletes or just deaf athletes? line 26 and 34 clarify that you are discussing the differences.

86 - starts with an apostrophe 

110 - there are an excessive amount of parenthesis in section 2.2

353 - remove "the"

Good contribution to the literature.

Author Response

(The authors gave the same response as above.)

Reviewer 3 Report

This paper is valuable as a report on the cardiorespiratory fitness and nutrition profile of deaf women's basketball team players. Although the sample size is small and there is a bias in the measurement method, it has important information that suggests that a nutritionally balanced diet will be needed to improve physical and athletic performance. Here are some comments to help authors publish successfully.

[Minor comments]

(Line 227-240; 247-248; 309-314) Where is the basis for the data described (related figures, tables or references)?

(Line 260-262) Isn't the maximum recommended amount of SFA 12% calories?

(Line 280-282) Is the copper missing?

(Table 2.) The average MFMI (3.0-3.9) is missing.

Author Response

(The authors gave the same response as above.)

Reviewer 4 Report

This article defines some performances variables and the diet of deaf women basketball players. The study is interesting because it offers new information. This topic still has to be deeply investigated in order to obtain better conclusions and more data to be compared.

I just have some concerns and questions for the authors regarding the paper.

I suggest changing the title of the article from "Cardiorespiratory Fitness and Diet Quality Profile of the Deaf Women’s Basketball Team Players" to "Cardiorespiratory Fitness and Diet Quality Profile of a Deaf Women’s Basketball Team Players" or to indicate that the study was performed on the Lithuanian team.

25. What do you mean by "low VO2peak (56-60ml/Kg/min)"?

61. during a basketball game.

98-99. Change to past sentence (i.e. take-took; prepare-prepared).

What do the numbers mean in the section "the content of the special work" in table 1? Please indicate.

195. the sentence "Significant differences were observed between the actual consumption of energy, macronutrients, and micronutrients and the recommended amounts" should not be here. Results maybe?

227-240 Much of the information provided here already appears in table 2. The data does not have to be presented twice, only once.

Author Response

(The authors gave the same response as above.)

Round 2

Reviewer 4 Report

The authors have addressed all of my concerns. I, therefore, suggest accepting it for publication. It has been an interesting read.